# Current and Potential Applications of Atmospheric Cold Plasma in the Food Industry

**DOI:** 10.3390/molecules28134903

**Published:** 2023-06-21

**Authors:** Darin Khumsupan, Shin-Ping Lin, Chang-Wei Hsieh, Shella Permatasari Santoso, Yu-Jou Chou, Kuan-Chen Hsieh, Hui-Wen Lin, Yuwen Ting, Kuan-Chen Cheng

**Affiliations:** 1Institute of Biotechnology, College of Bioresources and Agriculture, National Taiwan University, Taipei City 106319, Taiwan; d.khumsupan@gmail.com; 2School of Food Safety, Taipei Medical University, Taipei City 110, Taiwan; splin0330@tmu.edu.tw; 3Department of Food Science and Biotechnology, National Chung Hsing University, Taichung City 402, Taiwan; welson@nchu.edu.tw; 4Department of Chemical Engineering, Widya Mandala Catholic University, Surabaya 60114, Indonesia; 5Institute of Food Science and Technology, College of Bioresources and Agriculture, National Taiwan University, Taipei City 106319, Taiwan; lucy840218@gmail.com (Y.-J.C.);; 6Department of Optometry, Asia University, Taichung City 41354, Taiwan; 7Department of Medical Research, China Medical University Hospital, Taichung City 404327, Taiwan

**Keywords:** atmospheric cold plasma, food modification, active packaging, microbial inactivation, enzyme inactivation, food waste processing

## Abstract

The cost-effectiveness and high efficiency of atmospheric cold plasma (ACP) incentivise researchers to explore its potentials within the food industry. Presently, the destructive nature of this nonthermal technology can be utilised to inactivate foodborne pathogens, enzymatic ripening, food allergens, and pesticides. However, by adjusting its parameters, ACP can also be employed in other novel applications including food modification, drying pre-treatment, nutrient extraction, active packaging, and food waste processing. Relevant studies were conducted to investigate the impacts of ACP and posit that reactive oxygen and nitrogen species (RONS) play the principal roles in achieving the set objectives. In this review article, operations of ACP to achieve desired results are discussed. Moreover, the recent progress of ACP in food processing and safety within the past decade is summarised while current challenges as well as its future outlook are proposed.

## 1. Introduction

Incidents of foodborne outbreaks and subsequent recalls of food products are frequently caused by ineffective disinfection methods. In addition to concerns of microorganisms’ resistance to standard food processing practices, producers also need to consider other marketing factors, such as chemical-free products, minimal processing, and safety, to satisfy consumers’ demands [1,2,3]. Conventionally, thermal processes which involve the incorporation of heat, including pasteurisation and sterilisation, have been extensively exploited. However, the techniques have several drawbacks: long processing time, loss of sensory properties, and degradation of thermally sensitive nutrients. Scientists have looked into other approaches to overcome these problems.

Plasma is commonly referred to as the fourth state of matter based on the levels of energy, after solid, liquid, and gas. It comprises electrons, ions, neutral species, photons, and metastable and other excited gaseous atoms. Both man-made and naturally occurring plasmas can have a wide range of temperatures and densities. The ability to regulate their behaviour is of great interest to the scientific community [4,5,6]. The first application of plasma was conducted by Werner von Siemens in 1857 when he created a dielectric barrier discharge ozoniser to treat water. Then, the term plasma was later coined in 1928 by Irving Langmuir [7,8]. Plasma can be classified into two types: thermal and nonthermal (low temperature) [3]. Non-equilibrium or “cold” plasma, which are both types of nonthermal plasma, are states that are not in local thermodynamic equilibrium (usually less than 60 °C) [9]. 

As a result, cold plasma at atmospheric pressure has attracted substantial attention within the food industry in the past decade since the expensive vacuum pump can be omitted while the generated cold plasma still maintains similar properties. In food technology, a diverse application of atmospheric cold plasma (ACP) can be observed since it offers numerous benefits, as summarised in Figure 1. ACP is considered as an economical processing technology as it does not require heat, pressure, water, or additional chemical solvents [10]. Moreover, it utilises less energy than conventional methods due to shorter treatment time [3]. However, ACP still encounters some constraints that limit its full potential within the food processing industry. This review briefly provides the mechanism of how ACP is generated and summarises specific areas within the food industry where ACP can be employed. Furthermore, challenges of ACP in the field are discussed.

## 2. Mechanism of ACP

ACP is commonly generated at ambient temperature and pressure by the interaction between an electric field (usually a pair of electrodes) and gas molecules, which subsequently form charge carriers (a mixture of electrons and ions). Then, the free charge carriers are further excited by the electric field and collide with atoms and molecules in the gas or with electrode surfaces, producing a large quantity of new charged particles. A steady-state cold plasma at atmospheric pressure is formed when the particles and charge carrier losses are balanced [11]. Figure 2 shows a diagram of cold plasma generation at atmospheric pressure.

While the exact composition of plasma-generated species depends on various conditions such as gas type, voltage, and humidity, it typically contains reactive oxygen and nitrogen species (RONS), free radicals, excited molecules, and UV photons [3]. Within the food processing industry, some of the most common ACP generators are the dielectric barrier discharge (DBD), atmospheric plasma jet (APPJ), radio frequency (RF), spark and glow discharge, and gliding arc (GA). The general schematics of DBD and APPJ are illustrated in Figure 3. DBD offer certain advantages over other plasma devices, including its ability to generate high-energy species with minimal energy input. In addition, the potential to scale up DBD is greatly feasible due to its simple geometric design [12]. During the treatment of food products, DBD can be used to treat the sample directly or indirectly. The direct treatment involves positioning the food between the two electrodes, whereas the indirect treatment requires a container or a medium to contact the plasma discharge [13]. However, since plasma particles produced by DBD remain between two parallel electrodes, the sample shape and size could pose restrictions. Unlike DBD, APPJ releases the reactive species into an open environment which can be directly applied to the object without limitation to its size [14]. In radio frequency discharge, pulsed electric current is used to generate a plasma column in the centre of an electric coil. This plasma column produces a high-density ionized gas which can be utilized in various applications [15]. Spark discharge has the advantage of producing a remarkably high energy density, but its downside is the discharge short duration, which may lead to damage in both the electrode and material used [16]. The power of glow discharge is limited by the transition from glow to arc discharge. During this transition, the discharge voltage decreases, requiring an increase in current to maintain power at the same level, which in turn leads to gas heating [17]. GA discharges are relatively simple in structure and can operate under a variety of conditions; however, its drawbacks include limited gas–plasma interaction time and a low percentage of treated gas after processing [18].Therefore, each plasma device has its own benefits and drawbacks, which stipulates a thorough consideration in regard to the purposes of plasma treatment, surroundings, and target sample.

## 3. Atmospheric Cold Plasma (ACP) in Food Technology

### 3.1. Microbial Inactivation

Studies of ACP have reported promising results in the inactivation of pathogens in food processing. Table 1 is a list of selected publications related to the use of ACP to inactivate microorganisms.

Most studies focus on bacteria such as *Escherichia coli*, *Listeria monocytogenes*, and *Salmonella* species, which are some of the most common food pathogens. *Escherichia coli* is a Gram-negative bacterium commonly found in the gut of humans and animals. While most strains of *E. coli* are benign, some can be pathogenic and trigger severe diarrhoea. Each year, more than 1.7 billion people suffer from severe diarrhoea; in addition, approximately 760,000 children under 5 years old die from diarrheal diseases every year, making it the second leading cause of death amongst children under 5 years old [27]. Similarly, another rod-shaped, Gram-negative bacterium that has always been a major public health concern is *Salmonella*. *Salmonella* accounts for 155,000 deaths annually and it is one of the main causes for gastroenteritis [28]. Both *E. coli* and *Salmonella* can live in a variety of food products including meat, dairy, and vegetables [27,28].

Moreover, *Listeria monocytogenes* is another bacterial pathogen which results in 19% of the deaths of Americans each year in cases related to food contamination [29]. The growing popularity of ready-to-eat products as well as the refrigeration system in industrial-scale food processes make the bacteria such a prevalent pathogen as they can develop resistance to environmental stress and multiply in cold temperatures [30]. The research of ACP on tender coconut water concluded that ACP can be utilised as a nonthermal process to inactivate Gram-negative bacteria *E. coli* and Gram-positive bacteria *L. monocytogenes*, lengthening the shelf life of tender coconut water by up to 48 days [22]. The experiment was carried out using dry and modified air M65 (65% O_2_, 30% CO_2_, and 5% N_2_) with the treatment time at 120 s at 90 kV. According to the results from optical emission spectroscopy, RONS were responsible for bacterial cell leakage or induced other morphological changes in bacteria [22].

Overall, the mechanism of ACP to inactivate bacteria was examined and it was concluded that in Gram-negative bacteria, RONS generated by ACP damage the lipoproteins and peptidoglycans of the cell envelope which lead to cell leakage and disruption. Meanwhile, ACP does not promote cell leakage in Gram-positive bacteria but instead impair essential cellular components such as DNA [31].

ACP has also been extended to the inactivation of other microbes, including viruses, yeasts, and fungi. Although viral contamination in food is less frequent than that of bacteria, its effect on public health is as severe. Human norovirus is one of the viral pathogens which can also cause gastroenteritis. As the virus usually exists in water, it can pose danger to any food products that carry water, such as fruits, vegetables, and shellfish. Currently, human norovirus cell culture systems cannot be cultivated in a laboratory setting and the researchers have to rely on murine norovirus and Tulane virus as surrogates [32]. A viral inactivation experiment was performed on the Tulane virus and murine norovirus on blueberries using a plasma jet under the following conditions: 4 cubic feet/min (cfm), 0–60 s treatment time, and 7.5 cm treatment distance. The study observed a notable reduction of Tulane virus (1.5 PFU/g) compared to the control after 45 s of treatment time, which demonstrated the potential of ACP on blueberry processing [24]. In viruses, RONS are the main contributors of viral inactivation as they penetrate through capsid by diffusion and cause damage to the RNA [33].

The research on other pathogens was investigated. The effect of ACP on a yeast strain *Saccharomyces cerevisiae* was studied and it was reported that this medium can also influence the sterilisation efficacy in addition to plasma conditions (oxygen gas level, electrical power, and treatment time). According to the results, ACP showed the highest inactivation efficiency when *S. cerevisiae* is in water and saline solution. This outcome elucidated that ACP may be the most suitable food processing choice when the media are water or other biological fluids containing NaCl. Contrastingly, the plasma treatment of *S. cerevisiae* in YPD media exhibited the least efficiency, which asserts that the ability of ACP to inactivate microorganisms is reduced in nutrient-rich solutions [34]. Moreover, the study also concluded that OH radicals played the most vital role in yeast cell inactivation. The experiment assessed the level of oxygen reactive species and reported that the water media contained the highest level of OH radicals, whereas YPD had the lowest. One possible explanation is that oxygen radicals were quenched when ACP were treated in YPD solution, which resulted in less lipid peroxidation and subsequently less cell damage [34].

Aflatoxins, produced by the fungi *Aspergillus flavus* and *Aspergillus parasiticus*, are mutagenic and carcinogenic compounds discovered in 1960 during an outbreak called “Turkey ‘X’ Disease” in the UK [35]. B1 and B2 aflatoxins are produced by *A. flavus*, whereas B1, B2, G1 and G2 aflatoxins are all produced by *A. parasiticus* [26]. The most toxic aflatoxin is B1 as it is linked to liver cancer. All aflatoxins can commonly be found in poorly dried foods such as cereals, spices, and nuts, where the fungi can proliferate. Since aflatoxins are extremely tolerant to both heat and freezing temperatures, they can exist in food indefinitely [35]. Devi and team [26] published the results on the effect of ACP on fungal growth in groundnuts and elimination of aflatoxins. After the inoculated groundnuts were plasma-treated with 60 W for 15 min, they observed the reduction in *A. parasiticus* by 97.9% and that of *A. flavus* by 99.3%. The results from electron microscopy showed that RONS generated electroporation and etching of fungal spore membranes. This study indicated the potential application of ACP to eliminate pathogenic fungi and their toxins. Despite these diverse scientific studies, it should be noted that the efficacy of ACP depends on the microbial species, density of pathogens, plasma system, and treatment conditions [31].

### 3.2. Active Food Packaging

Packaging can improve food quality and safety by protecting the product from undesirable external conditions such as moisture, microorganisms, dust, and extraneous materials. However, traditional packaging is limited in extending the shelf life of food; hence a functional packaging, also known as active packaging, is explored. According to the statement issued by Commission Regulation (EC) No. 450/2009, the active packaging aims to “deliberately incorporate components that would release or absorb substances into or from the packaged food or the environment surrounding the food” [36]. In studies of plasma application on packaging film, the examined conditions typically include plasma treatment conditions as well as physicochemical properties of the modified film, as shown in Table 2.

Wong et al. [37] investigated the effectiveness of ACP-treated PE film coated with chitosan and gallic acid film for tilapia fillet preservation. The addition of 1% chitosan and gallic acid extended freshness of the fillet for 14 days as it can hinder bacteria accumulation by 1.52 log CFU/g compared to control, and delayed volatile basic nitrogen and thiobarbituric acid by 89.9% and 33.3%, respectively [37]. At the molecular level, the bombardment of reactive species, particularly OH radicals, oxidises covalent bonds on the film surface to form carboxyl and other oxygen-containing functional groups [38]. This process also enhances the roughness of the film surface which can facilitate the incorporation of other antioxidant or antimicrobial compounds [39,40]. Thus, it can be perceived that ACP has the potential to be integrated into many steps within the food packaging process, from the production of active packaging to post-contamination prevention. The ability to control the production of OH radicals and other reactive species would certainly be beneficial to the progress in this technology. 

### 3.3. Food Allergen Mitigation

Food allergy is *a critical* food safety issue, and its prevalence is growing continuously between 2 and 10% [41]. Many studies investigate the influence of ACP on food allergens, which are listed in Table 3. The eight major allergens, such as milk, eggs, fish, shellfish, tree nuts, peanuts, wheat, and soybean, account for 90% of food allergies and serious allergic reactions in the world [42]. While thermal processing is commonly used to reduce allergenicity in food, most allergenic proteins are thermally stable [43].

ACP has become an alternative approach to deactivating allergens via structural changes in proteins. Ng et al. [44] treated milk allergens casein, β-lactoglobulin, and α-lactalbumin through spark discharge (SD) and glow discharge (GD) ACP. After SD-ACP and GD-ACP treatment for 30 min, the antigenicity of casein was decreased by 49.9 and 91.1%, whereas that of α-lactalbumin was reduced by 49.5 and 45.5% compared to control, respectively. In milk, RONS from ACP decrease the antigenicity by denaturing α-lactalbumin and β-lactoglobulin proteins. This is achieved by the destruction of hydrogen bonds, which stimulates protein aggregation through disulphide bonding [50]. Essentially, ACP induces intermolecular cross-linkage from the cysteine residues of milk allergens, which affects primary and secondary structures and hence their binding capacity to antibodies [45].

A similar mechanism also occurs in the egg allergen. In more than 35% of patients who are allergic to eggs, lysozyme is the main cause of their allergic reactions [51]. Lysozyme has been used extensively as a food additive as it can hydrolyse the cell wall of Gram-positive bacteria and induce cell disruption, increasing products’ shelf life. Consequently, ingredient labels must add the information about any egg-derived additives including lysozyme for safety purposes. Using a low-frequency plasma jet on a 0.3 mL lysozyme sample containing 0.1 mg/mL in 10 mM phosphate buffer (pH 7.4), a denaturation of lysozyme could be detected [52]. It was proposed that the denaturation mechanism of lysozyme by ACP was attributed to the RONS, which induce chemical changes in certain amino acids such as cysteine, phenylalanine, tyrosine, and tryptophan. After the exposure to ACP, the allergenic proteins are reported to have lost their secondary structures, such as α-helices and β-sheets, which result in the destruction of enzymes’ binding sites [52]. This study highlights the effect of ACP on lysozyme’s activity, which is caused by structural changes of the protein. Nevertheless, it is also possible for new proteins to be formed after the interaction with active species [31].

Peanut allergy is the most prevalent food allergy in many Western countries. The effects of ACP (60 min of treatment time) on major peanut allergens Ara h 1 and Ara h 2 were observed using whole peanut and defatted peanut flour, which showed that the antigenicity was reduced by 65% for Ara h 1 and 66% for Ara h 2. The decrease in antigenicity may be due to the ability of plasma reactive species to change secondary structures of the allergens, which also reduces peanut protein solubility [46]. Liu et al. [48] indicated that the conformational alteration is caused by an oxidation of peptide bond amino groups, such as Trp, Tyr, and Phe amino acid residues. Furthermore, the cleavage of these polypeptide chains can partially diminish linear epitopes. It is intriguing to note that moderate ACP treatment improves the functionality of soy protein, such as solubility, emulsification, and foaming properties, while overexposure may result in denaturation of the soy protein [48,49].

### 3.4. Enzyme Inactivation

An ability to regulate enzymes of food products can positively contribute to their preservation and storage. Table 4 presents the publications of ACP in enzyme inactivation.

Studies of endogenous food enzyme inactivation have been investigated using DBD and a plasma jet to increase shelf life. One such food is wheatgerm. After treating wheatgerm at 24 kV for 25 min, the results showed that lipase decreased by 25.03% while lipoxygenase dropped by 49.98%. However, the extension of treatment time beyond 25 min did not drastically improve the inactivation efficiency. It is also interesting to note that the inactivation effect was not permanent, as the both lipase and lipoxygenase enzymes recovered some of their activities during the storage period. However, the result showed that ACP did not affect the phenolic content of wheatgerm. This suggests that ACP can be a great boon when it comes to inactivation of endogenous enzymes in food processing [53].

**Table 4 molecules-28-04903-t004:** Selected publications of ACP on enzyme inactivation in food products.

Enzyme Inactivation									
Food Product	Plasma Device	Enzymes	Reduction of Enzyme Activity	Parameters					Reference
				Exposure Time (s)	Exposure Distance (mm)	Input Power (W)	Voltage (kV)	Frequency	
White mushroom	DBD	β-1,3-glucanase, MDA, and PPO	46.2%; 47.5%; 42.0%	60	100	30	-	13.6 MHz	[7]
Mushroom (*Agaricus bisporus*)	DBD	PPO	70.0%	600	38	-	50	-	[54]
Milk	DBD	ALP	50.0%	120	40	-	60	50.0 Hz	[55]
Wheatgerm	DBD	Lipase and lipoxygenase	25.0%; 50.0%	1500	20	-	24	50.0 Hz	[53]
Hen egg white	DBD	Lysozyme	50.0%	720	3	-	0.14	16.0 kHz	[56]
Hen egg white	Plasma jet	Lysozyme	60.0%	720	6	-	0.08	24.0 kHz	[56]
Fresh-cut melon	DBD	POD and PME	17.0%; 7.0%	900	5	-	15	12.5 kHz	[57]
Bananas	DBD	POD and PPO	64.4%; 62.6%	120	6	-	0.040	10.0 kHz	[58]

Other endogenous enzyme inactivations are studied, including pectin methylesterase (PME), polyphenol oxidase (PPO), and peroxidase (POD). All of them are normally found in fruits and vegetables as they are responsible for ripening and softening [59,60]. The presence of these enzymes in food commodities can shorten shelf life, which depreciates their market values. Typically, pasteurisation is implemented in the food process to inactivate these enzymes as well as other microorganisms [60].

Tappi and team [57] used fresh-cut melon to observe the effect of ACP on POD and PME. According to the result, POD and PME were reduced by 17% and 7%, respectively, after 15 min of plasma exposure. After the treatment, the samples could be stored up to 4 days at 10 °C compared to 2.5–3 days of untreated samples [57]. The study highlights ACP’s ability to inactivate undesirable enzymes and microorganisms in food processing. Further developments of the method may be highly valuable to the food industry.

### 3.5. Food Drying Pre-Treatment

In order to lengthen food shelf life and reduce transportation costs, dehydration is the process that is commonly used. However, conventional drying techniques may cause degradation to heat-sensitive compounds which results in quality deterioration, such as loss of texture, nutrients, pigment, and aroma. For this reason, there has been an increasing interest in the use of nonthermal processing which aims to enhance the drying process. Table 5 shows the publications of ACP in assisted drying.

The drying processing is a major operation, especially in agro products. Conventionally, a pre-treatment method, such as dipping in an alkaline solution, is employed to hasten the drying step and extend the shelf life, but concerns regarding chemical wastewater and toxicity from chemical residues are raised. Using a plasma jet, Huang and colleagues [61] pre-treated the grape surface three times at a power of 500 W and a frequency of 25 kHz to observe the drying rate of plasma-treated grapes. The results showed that the rate of moisture loss increased as the distance between the plasma nozzle and grape decreased. The experiment reported that no changes in appearance, colour, and antioxidant content of the sample were detected after the ACP treatment [61]. Moreover, they also detected an increase in the total phenolic content (TPC) from approximately 30 mg to 60 mg per 100 g of raisin, in addition to the change in antioxidant capacity from 4.5% to 10%. This is due to the efficiency of moisture diffusivity which reduces the drying time and energy consumption by up to 26.27 and 26.30%, respectively [61]. Similarly, in wolfberry, 45 s of ACP treatment could also shorten the drying time by 50% and increase the rehydration ratio by 7–16%. As the detection of phytochemical contents increased after the ACP treatment, the authors speculated that ultrastructure alteration results in the release of compounds that were trapped within cells, thereby raising the phytochemical contents [68].

Corn is one of the major grains in the world. Fresh corn kernels are easily spoiled due to their high moisture content, which means that drying technology is a crucial operation in corn post-harvest handling to extend the storage period [63]. ACP pre-treatment can be implemented to improve the drying efficiency of corn kernels. Setting the parameters at 500 W for 30 s, the drying time was reduced by 21.52%, and the drying rate was increased by 8.15%. The activation energy of drying kinetics from ACP was 47.79 kJ mol^−1^, compared to 54.82 kJ mol^−1^ of the control group. Furthermore, atomic force microscopy displayed the surface topography of plasma-treated corn kernel having shrunk or damaged granules [63,70]. This result may help explain the high effectiveness of ACP in the drying pre-treatment process.

### 3.6. Pesticide Decontamination

The utilisation of pesticides in crops contributes to higher production yields; however, the chemical residues left in food after cultivation create safety concerns as they are highly toxic to consumers [71]. To optimise the pesticide decontamination process, many scientific reports propose the implementation of ACP to degrade pesticides in food crops, as shown in Table 6.

One of the most common pesticides in the world is an organophosphate chemical called chlorpyrifos. Having been introduced in 1965, chlorpyrifos is widely used in the cultivation of various plants, such as fruits, vegetables, nuts, and grains [72]. Consequently, many food crops are found to contain high concentrations of this pesticide. The publication of chlorpyrifos and carbaryl degradation on the corn surface using ACP was observed. Based on the optimisation study, the treatment time of 60 s, air flow rate of 1000 mL/min, power of 20 W, and frequency of 1200 Hz resulted in 86.2% and 66.6% degradation efficiency of chlorpyrifos and carbaryl, respectively. Moreover, the treatment did not significantly affect the nutritional quality of corn, showcasing that ACP can be a promising food processing treatment for pesticide degradation [73]. 

Another study of chlorpyrifos was conducted along with a pesticide called diazinon on apples and cucumbers. The pesticide-dipped fruits were used to test the efficiency of ACP on the toxic degradation in which the humidity, firmness, colour, and sugar percentage of the fruits were determined. According to the results, ACP had relatively minimal effects on those parameters when it was set at 10 min exposure and 13 kV; however, the changes occurred when the treatment time increased [74]. Another interesting point should be mentioned, since the study illustrated varying detoxification efficiency between halogenated pesticides such as chlorpyrifos and the non-halogenated ones such as diazinon. Due to the chemical composition, polarity, or penetration of plant tissue, ACP was able to remove dianizon more effectively than chlorpyrifos [74]. According to the study, ACP may be highly applicable in food processing as a pesticide decontamination method.

**Table 6 molecules-28-04903-t006:** Selected publications of ACP on pesticide decontamination in food products.

Pesticide Decontamination									
Food Product	Plasma Device	Pesticide Active Ingredients	Degradation of Pesticides	Parameters					Reference
				Time (s)	Distance (mm)	Input Power (W)	Voltage (kV)	Frequency	
Strawberries	DBD	Azoxystrobin, cyprodinil, fludioxonil, and pyriproxyfen	69%; 45%; 71%; 46%	300	40	-	80	50.0 Hz	[75]
Blueberries	DBD	Boscalid and imidacloprid	80.2%; 75.6%	300	40	-	80	50.0 Hz	[76]
Corn	DBD	Chlorpyrifos and carbaryl	86.2%; 66.6%	60	6	20	-	12.0 kHz	[73]
Mango	Gliding arc	Chlorpyrifos and cypermethrin	74%; 62.9%	300	2.5	600	8	-	[77]
Apple	DBD	Chlorpyrifos and diazinon	87.0%; 87.4%	600	7	-	13	13.0 kHz	[74]
Cucumber	DBD	Chlorpyrifos and diazinon	33.7%; 82.2%	600	7	-	13	13.0 kHz	[74]
Lettuce	DBD	Chlorpyrifos and malathion	51.4%; 53.1%	120	35	-	80	50.0 Hz	[78]

### 3.7. Food Modification

Plasma treatment has a potential to modify food products to enhance properties or nutrition. Some studies of ACP and its application to alter characteristics of food are listed in Table 7. Cold plasma can be used in starches to decrease viscosity, molecular weight, and gelatinisation temperatures [79].

Corn starch is one of the food materials that can be altered to improve its poor physical and functional properties, enhancing its solubility or viscosity. A study reported physicochemical changes in corn starch after being treated with ACP for 30 min at 400–800 W. After washing with distilled water, the peak viscosity, final viscosity, and setback of starch samples were reduced by 87.1%, 92%, and 93.3%, respectively. The results highlight that ACP causes etching on the starch grains, which contributes to solubility and clarity [80].

Wheat is one of the most common staple crops in the world as it is used as an ingredient in bread, pasta, and other bakery products. Consequently, numerous chemicals and enzymes are used on wheat as oxidising or bleaching agents. To avoid potential toxicity from these additives, the effect on wheat flour of nonthermal technology, such as high pressure processing and ACP, has been widely studied. Since ACP can generate strong oxidising agents, such as RONS, the technique can replace the conventional oxidising agent without leaving any toxic residues during wheat processing. According to the report on ACP on wheat flour, treatment times ranging between 5 and 30 min at 80 kV induce depolymerisation of starch and reduces its crystallinity, which essentially increases the hydration and viscosity of wheat flour [81].

**Table 7 molecules-28-04903-t007:** Research studies of ACP on food modification.

Food Modification									
Food Product	Plasma Device	Modification	Results	Parameters					Reference
				Time (s)	Distance (mm)	Input Power (W)	Voltage (kV)	Frequency	
Fenugreek	DBD	Galactomannand yield	122%	1800	40	-	80	60 Hz	[82]
Maize	DBD	Increase in crystallinity	36.90%	600	5	-	0.138	50 Hz	[83]
Wheat	DBD	Increase in viscosity	17.60%	1800	30	-	80	50 Hz	[81]
Whey protein isolate	DBD	Emulsification enhancement	25.00%	300	44	-	70	-	[84]
Xanthan gum	SBD	Increase in viscosity	40.00%	1800	53	250	3.5	15 kHz	[85]
Pomegranate juice	Plasma jet	Increase in phenolic compounds	33.00%	300	22	6	2.5	25 kHz	[86]
White grapes	Plasma jet	Drying speed	20.00%	36,000	10	500	-	25 kHz	[61]
Chili pepper	Gliding arc	Drying speed	16.70%	30	60	750	-	20 kHz	[64]
Wolfberry	Gliding arc	Drying speed	14.10%	60	60	750	-	20 kHz	[68]

Similar results could also be observed in xanthan gum. While various chemical and enzymatic techniques to improve its functionality exist, the treatments can be costly or involve tedious procedures. Using ACP treatment (60 W for 20 min), Bulbul et al. [87] found an increase in the porosity and compressibility index of xanthan gum. Moreover, another study demonstrated similar results after exposing xanthan gum to ACP for 20 min at 3.5 kV. The samples showed lower shear viscosity and increasing emulsifying capacity without any effect on their whiteness [85]. The research investigation concluded that ACP can be a practical processing technique for xanthan gum to expand its functional characteristics.

Under the scope of protein modification, bovine serum albumin (BSA) was treated with ACP, which caused protein unfolding and changes in the secondary structure. This finding suggests that ACP promotes the structural alteration, aggregation, peptide cleavage, and side-group modification of proteins [88]. Based on the characterisation study, ACP promotes structural conformation and unfolding of the polypeptide chain, which leads to more hydrogen bonding [89]. To summarise, while many researchers study the ability of ACP to alter polysaccharide and protein structures [90,91,92,93], few publications actually evaluate its exact mechanism, particularly on the physicochemical reactions between proteins and active species. Thus, more research on the topic is vital to gain better insights on its efficacy.

### 3.8. Nutrient Extraction

Polyphenols are antioxidants present in fruits and vegetables, which can help prevent lipid oxidation and provide the colour and flavour in food. Table 8 shows selected publications on the effects of ACP on nutrient extraction.

Many researchers claim that ACP can increase total phenolic content (TPC) in white grapes [61], pomegranate juice [86], blueberry juice [94], cashew apple juice [99], tomato pomace [95], and dry peppermint [100]. Nevertheless, operation factors, such as input power, voltage, treatment time, gas, frequency, and gas flow rate, play important roles in the treatment. The working gas is one of the most critical elements since it determines the compounds that are activated. In the case of tomato pomace, the exposure of atmospheric pressure plasma coupled with helium or argon resulted in a significant increment of antioxidant capacity, TPC, quercetin, and naringenin [94]. Argon plasma pre-treatment is shown to improve tomato pomace’s extraction rate, TPC, antioxidant capacity, and flavonoids [95], whereas helium is more suitable for enhancing those variables for grape pomace [96].

Nevertheless, not all studies on plasma report the increase in TPC. Setting the voltage at 80 kV, a reduction in total phenolic (from 720.62 to 445.02 gallic acid equivalent µg/mL) and flavonoid (from 265.21 to 211.46 catechin equivalent µg/mL) contents was recorded in grape juice when the exposure time increased to 4 min [101]. Moreover, the gas flow rate can also affect vitamin C, TPC, flavonoids, and antioxidant activity due to the interaction between phenol compounds and RONS [99,102,103]. As a result, the chemical structure of vitamin C may be degraded when the plasma exposure is too high [99]. 

According to these publications, ACP is shown to help improve concentration, colour, sensory properties, and nutrient retention. This can be justified by the ability of the cold plasma to disrupt the surface of cell membranes and release bioactive compounds within the cells [104]. Many research publications assert that the physical effect from low ion bombardment and chemical effects from free radicals, ions, and other plasma substances of ACP promote changes on the sample’s surface. One of hypotheses that may explain the higher content of polyphenols after ACP processing is the rupture of cell membranes, which leads to the release of potent compounds [105]. Nevertheless, comprehensive research on plasma extraction technology is limited, as most publications focus on polyphenol extraction and enhancement of antioxidant activity [106,107]. Unlike other processing technologies, ACP is not directly utilised as an extraction method; rather, it is commonly used as a pre-treatment step [108]. It can thus be concluded that the effects of ACP on each sample vary depending on the gas, plasma type, treatment time, flow rate, and energy.

### 3.9. Food Waste Processing

The accumulation of food waste at landfill sites instigates the need to efficiently manage it. Research studies have come up with ways to valorise different types of food waste using ACP. Table 9 lists scientific publications which study the results of ACP to enhance the production of economically valuable compounds from food waste.

Successful ethanol production from sugarcane bagasse was studied, whereby ACP was utilised to detoxify inhibiting compounds in the pre-treatment process. Lin and team [109] reported that ACP set at 200 W for 25 min on sugarcane bagasse hydrolysate can remove as much as 31% of formic acid, 45% of acetic acid, 100% of furfural, and 81% of hydroxymethylfurfural (HMF), enhancing ethanol productivity from 0.25 to 0.65 g/L/h. Maintaining the same parameters of ACP, they could also use the carbon source obtained from the ACP-treated sugarcane bagasse hydrolysate to produce bacterial cellulose [110]. The study showed the high efficacy of ACP to eliminate toxic compounds, indicating that ACP can serve as a possible processing alternative in the pre-treatment step to help lower the cost.

**Table 9 molecules-28-04903-t009:** Selected publications of ACP on food waste processing.

Food Waste Processing										
Food Waste	Plasma Device	Products	Results	Parameters						Reference
				Gas Type	Time (s)	Distance (mm)	Input Power (W)	Voltage (V)	Frequency (Hz)	
Grape pomace	DBD	Phenolic compounds	22.8%	Air	900	52	-	120	60	[96]
Pineapple peel	DBD	Bacterial cellulose	3.82 g/L	Ar, Air	900	10	600	-	-	[111]
Sugarcane bagasse	Plasma jet	Bioethanol production	38.5%	Ar	1500	10	80–200	-	-	[109]
		Bacterial cellulose	1.68 g/L	Ar	1500	10	200	-	-	[110]
Wheat straw	DBD	Methane	45.0%	Air	3600	20	230	-	10 kHz	[112]

Likewise, agro-industrial waste such as pineapple peel can be valorised to reduce landfill sites. In this study, pineapple peel waste is hydrolysed for bacterial cellulose production. The results illustrated that even though ACP with argon plasma at the power of 80–200 W possesses higher ability to remove the toxic compounds (such as formic acid, furfural, and HMF) in the pineapple peel waste hydrolysate than air plasma at the power of 500–600 W, it also contributed to relatively more sugar degradation. Since the study concluded that both ACP-treated and untreated hydrolysate can be fermented without jeopardising the quality of bacterial cellulose, it emphasises how pineapple peel waste can be one of the renewable sources for bacterial cellulose production [111].

In conclusion, ACP illustrates high potential in the area of food waste processing, as it enables the waste to be repurposed as renewable materials, reinforcing the concept of sustainability. Nonetheless, more research studies are essential to diversify the application of ACP in this particular sector.

## 4. Challenges of ACP in Food Industry

Despite a plethora of uses in numerous industries, ACP still encounters certain challenges which obstruct its applicability. In food processing, treatment time is undoubtedly one of the most critical parameters as it directly contributes to effectiveness. While an optimised time can improve functionality or enhance desired results, overexposure may lead to product degradation. When ACP is used to degrade allergens in soybean protein isolate, for instance, overtreatment results in the aggregation of protein, which leads to the loss of its functionality [49]. 

Moreover, different gas types and flow rates may substantially contribute to the treatment efficacy of specific compounds. Argon plasma, for example, may be a more efficient working gas than N_2_, O_2_, and air plasma when extracting essential oil from lemon peel [113,114]. On the other hand, a low flow of N_2_ on ACP can improve total polyphenol content in cashew apple juice by activating existing antioxidant compounds as a response against the generated plasma reactive species, but the higher gas flow may result in degradation of these phenolic compounds [99]. In consequence, the objective of plasma treatment should be considered before standard parameters can be customised for each food product.

Beyond the conditions, the type of ACP generator also contributes to the challenges. In the study of milk allergenicity, Ng and colleagues [44] discovered that glow and spark discharge ACP shows dissimilar efficacy on the types of protein (casein, β-lactoglobulin and α-lactalbumin) after using the same treatment conditions. Therefore, in addition to identifying the type of ACP devices, researchers are also required to carefully characterise optimal conditions for each plasma source. 

Additionally, instrument size creates some concerns. As the current dimensions are built for research purposes, the device may not be suitable in an industrial operation which requires continuous processing in large bulk. Thus, the upscale technology of ACP is of paramount significance as it may help create a standard of procedure within the food industry [10,79].

Furthermore, very few researchers focus on the toxicological effects of ACP on food. Although ACP can potentially replace conventional food processing techniques, limited information on the consumption of ACP-treated foods is documented. Hence, additional studies on the chemistry of ACP are crucial to ensure that potential by-products from the process do not pose any health risk for consumers. To the best of our knowledge, a formal policy to regulate the operation of ACP on food has yet to be established by the FDA or other food agencies. If all of these concerns are addressed, ACP could be integrated to enhance the processing steps and safety in the food industry.

## 5. Conclusions and Future Outlook

The integration of ACP in the food sector shows promising results as the method yields high productivity in inactivating food pathogens and enzymes, decontaminating pesticides, improving food modification techniques, as well as enhancing the processing of food waste. With such attractive features, the need to shed light on the chemistry of these operations is more urgent than ever. Once the insights on the chemical interactions of ACP are well-studied, the processing standard can then be formulated to make sure that plasma-treated foods are adequately safe for human consumption. Therefore, the scaled-up cold plasma system may contribute to the treatment quality by improving processing speed and continuity. This may lead to an explosive progress as more researchers may attempt to adapt this technology in other areas. In consequence, each laboratory would be able to compare results and devise standards where food and other materials can be effectively processed.

## Figures and Tables

**Figure 1 molecules-28-04903-f001:**
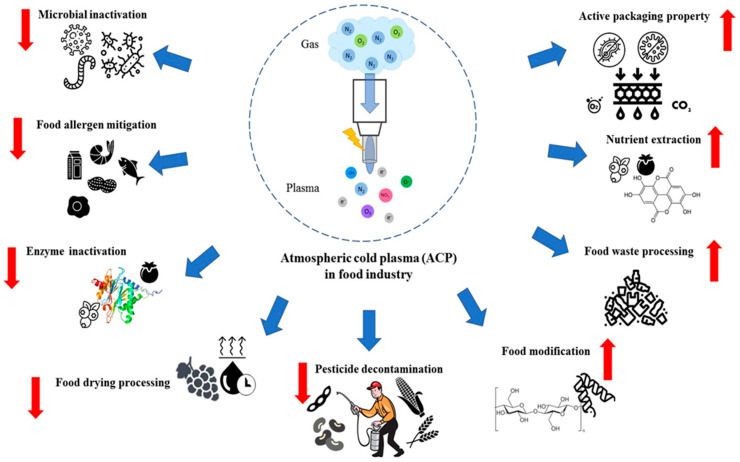
Applications of atmospheric cold plasma (ACP): Depending on the set parameters, ACP can be utilised to inactivate pathogens and compounds, or modify food products.

**Figure 2 molecules-28-04903-f002:**
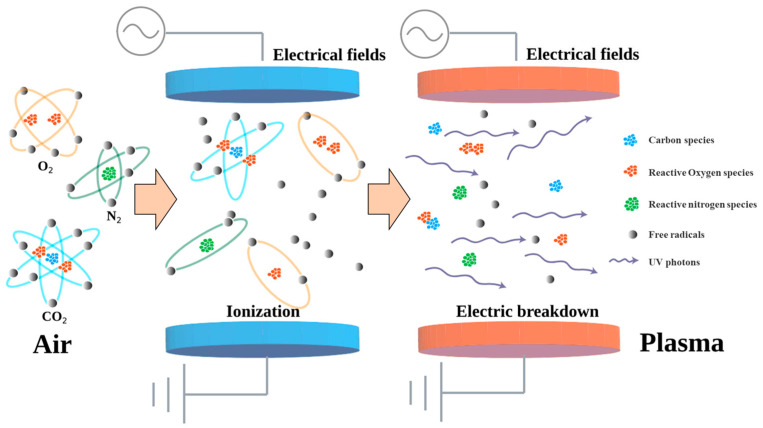
Generation of atmospheric cold plasma (ACP): Initially, gas molecules are ionised in an electrical field, producing a mixture of electrons and ions. Then these charged carriers are further excited and create a large quantity of new charged particles.

**Figure 3 molecules-28-04903-f003:**
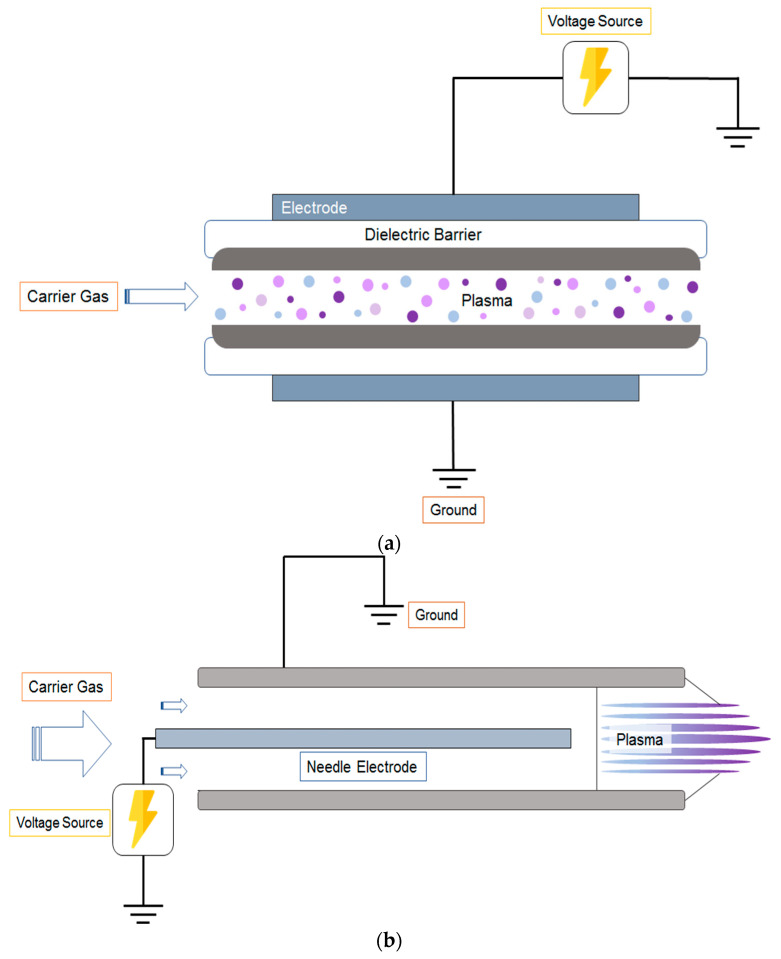
General schematics of plasma devices: (**a**) Dielectric barrier discharge (DBD): Plasma creates a uniform glow between the two parallel electrodes. (**b**) Atmospheric plasma jet (APPJ): Plasma reactive species exit through the nozzle into the open environment.

**Table 1 molecules-28-04903-t001:** Scientific studies on the inactivation of microorganisms by ACP.

Food Product	Plasma Device	Microbial Strains	Reduction (log CFU)	Parameters					Reference
				Exposure Time (s)	Exposure Distance (mm)	Input Power (W)	Treatment Voltage (kV)	Frequency	
Prepackaged mixed salad	DBD	*Salmonella*	0.8/g	180	30	-	35	1.1 kHz	[19]
Golden Delicious apples	DBD	*Salmonella* and *E. coli*	5.3/cm^2^; 5.5/cm^2^	240	35	200	-	50 Hz	[20]
Boiled chicken breast cubes	DBD	*Salmonella*, *E. coli*, *L. monocytogenes*, and *Tulane virus*	3.7/cube; 3.9/cube; 3.5/cube; 2.2 PFU/cube	210	12	-	39	-	[21]
Tender coconut water	DBD	*L. monocytogenes* and *E. coli*	2.0/mL; 2.2/mL	120	10	-	90	60 Hz	[22]
Tilapia fillet	DBD	*V. parahaemolyticus*	1.8/g	60	100	30	-	13.6 MHz	[1]
		*S. enteritis*, *L. monocytogenes*	2.34 log CFU/g; 1.69 log CFU/g	300	52	70	80	60	[23]
Blueberries	Plasma jet	Tulane virus and murine norovirus	3.5/g; 5.0/g	120; 90	75	549	-	47 kHz	[24]
Military rations snack	Plasma jet	*A. flavus*, yeast-mold, and aflatoxin	4.3/g; 4.6/g; 3.0/g	360	30	-	9	-	[25]
Groundnuts	DBD	*A. flavus*, *A. parasiticus*, and aflatoxin	1.2/g; 1.2/g; 0.3/g	720	30	60	2	-	[26]

**Table 2 molecules-28-04903-t002:** Effect of cold plasma treatment on active packaging in food.

Food Matrix	Plasma Device	Film Materials	Treatment Conditions			Physicochemical Change (Optimisation Methods Were Chosen)					Reference
			Input Power (W)	Treatment Time (s)	Frequency (MHz)	Thickness (mm)	Tensile Strength (MPa)	Elastic Modulus (Mpa)	Elongation at Break	Water Vapour Permeability	
Tilapia fillets	RF- plasma	CNMA- CMC/LDPE	30	60	13.56	+29.97%	+13.58%	-	-	-	[1]
Chicken breast fillets	-	1–3% SEO- CS/LDPE	84	10	-	+650%	−3%	-	−27%	−96.7%	[2]
Cooked turkey meat	Plasma Jet	Citrus/PET	-	-	30 kHz	+150%	-	-	-	-	[4]
Korean steamed rice cakes	DBD- plasma	Nylon/PP Nylon/LDPE	21 kW	180	-	-	Nylon/PP + 1.6% Nylon/PE − 0.5%	Nylon/PP + 1.2% Nylon/PE + 0.5%	Nylon/PP − 0.3% Nylon/PE + 0.9%	Nylon/PP − 6.25% Nylon/PE − 7.7%	[5]
Button mushroom (*Agaricus bisporus)*	RF- Plasma	CMC, COL/LDPE	30	60	13.56	-	+7.6%	+47.43%	-	+114%	[7]

RF—radio frequency; CS—chitosan; LDPE—low-density polyethylene; DBD—dielectric barrier discharge; PE—polyethylene; CNMA—cinnamaldehyde; CMC—carboxymethyl cellulose; COL—collagen; SEO—summer savoury essential oil; PET—polyethylene terephthalate; PP—polypropylene.

**Table 3 molecules-28-04903-t003:** Research studies of ACP on food allergen mitigation.

Food Allergen Mitigation									
Allergens	Plasma Device	Sample Types	Antibody Binding Capacity	Parameters					Reference
				Exposure Time (min)	Exposure Distance (mm)	Input Power	Input Voltage (kV)	Frequency	
Casein	Plasma jet (spark discharge)	Allergenic protein solution	↓ 49.9%	30	2.5	-	8	25 kHz	[44]
α-lactalbumin			↓ 49.5%						
β-lactoglobulin			↑ 250%	10					
Casein	Plasma jet (glow discharge)		↓ 91.1%	30	5	-	5	25 kHz	
α-lactalbumin			↓ 45.5%						
β-lactoglobulin			↑ 300%	10					
β-lactoglobulin	DBD	Allergenic protein solution	↓ 58.21%	4	-	-	40	12 kHz	[45]
Ara h 1	Plasma jet (pin-to-plate)	Whole peanut	↓ 39.32%	60	70	-	32	52 kHz	[46]
		Defatted peanut flour	↓ 65%						
Ara h 2		Whole peanut	↓ 46%						
		Defatted peanut flour	↓ 66%						
β-conglycinin (Gly m5)	Plasma jet	Soy protein isolate	↓ 89%	90	-	12 kW	-	2.45 GHz	[47]
Glycinin	DBD	Allergenic protein solution	↓ 91.64%	5	50		40	20 kHz	[48]
			↓ 81.49% *						
Soy allergens	DBD	Soy protein isolate	↓ 75% *	5	35	-	40	120 Hz	[49]

↑—increase; ↓— decrease; *—The binding capacity of IgE antibodies.

**Table 5 molecules-28-04903-t005:** Scientific publications on food drying pre-treatment by ACP.

Food Drying Processing									
Food Product	Plasma Device	Drying Temperature (°C)	Reduction of Drying Time	Parameters					Reference
				Exposure Time	Exposure Distance (mm)	Input Power (W)	Input Voltage (kV)	Frequency	
Grape	Plasma jet	70	20%	3 times	10	500	-	25 kHz	[61]
	Plasma jet	60	26.27%	50 s	35	300	27	50 Hz	[62]
Corn kernels	DBD	37.5	21.52%	30 s	-	500	-	40 kHz	[63]
Chili pepper	Plasma jet	70	~16.6%	30 s	60	750	-	20 kHz	[64]
Shiitake mushroom	Plasma jet	50, 60, 70	The higher drying rate at 50 and 60 °C.	60 s	50	650	-	-	[65]
	Plasma-activated water				-				
Tucumã	DBD	60	61.1%	10 min	15	-	20	200 Hz	[66]
Jujube	Plasma jet	70	12.08%	1 min	50	650	5	40 kHz	[67]
	Plasma-activated water		Non-effect	10 min	-				
Wolfberry	Plasma jet	65	50%	30 s	60	750	-	20 kHz	[68]
Saffron	Plasma jet	60	54.05%	60 s	-	1000	8	50 Hz	[69]

**Table 8 molecules-28-04903-t008:** Recent publications of ACP on nutrient extraction.

Nutrient Extraction									
Food Matrix	Plasma Device	Parameters						Results	Reference
		Input Power (W)	Voltage (kV)	Time (min)	Working Gas	Frequency (kHz)	Gas Flow Rate (L/min)		
White grapes	Jet plasma	500	-	3–7	Air	25.00	40.00	TPC and antioxidant capacity increased more than twofold	[61]
Blueberry juice	Jet plasma	-	11	-	Ar, O_2_	1.00	1.00	TPC increased by 7.34%, and antioxidant capacity increased	[94]
Tomato pomace	DBD plasma	-	60	15	Ar, He, N_2_, air	-	-	TPC increased by 24.07%, and antioxidant capacity by 30%	[95]
Grape pomace	DBD plasma	-	60	5–15	He	-	-	Increased the yield of phenolic extracts; improved antioxidant capacity	[96]
Fenugreek	DBD plasma	-	120 V	30	air	0.06	-	Increased the extraction yield of fenugreek galactomannan	[82]
Tomato	DBD-plasma	0.55–1.43	13–17	5–45	air	0.05	15.00	Increased the weight of tomato by 20–40%	[97]
Black gram	Jet plasma	-	3–6	3–15	O_2_	3.00–10.00	0.25	Chlorophyll content increased by 23.80% and total soluble protein and sugar concentrations increased by 33.28% and 51.73%, respectively	[98]

## Data Availability

Not applicable.

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
