# Peer review of "Current and Potential Applications of Atmospheric Cold Plasma in the Food Industry"

_molecules, 2023, doi:10.3390/molecules28134903_

Round 1

Reviewer 1 Report

The authors present a review on the recent progress of atmospheric cold plasma in food processing within the past decade. Current challenges as well as its future outlook are also presented. However, the description of the non-equilibrium plasmas as well as the plasma-based devices contain several errors and should be considerably improved before suggesting the manuscript for publication. Please note that a non-equilibrium or cold plasma is not a subcategory of non-thermal plasma. The terms non-thermal plasma, cold plasma and non-equilibrium plasma refer all to a state of plasma which is far from the local thermodynamic equilibrium. Particularly the non-thermal plasmas with gas temperatures below 60 °C are suitable for plasma medicine and biological applications (as food processing). In addition, the sentence ‘In order to generate cold plasma, a low-gas pressure system is required to minimize molecular collisions’ should be explained or removed. Also, the sentence ‘DBD offer certain advantages over other plasma devices: Principally, high energy species can be generated from DBD with as minimal energy as 1 eV’ is wrong, and should be rewritten or removed. The Section 2 Mechanism of ACP needs to be rewritten. Typical plasma values expected for the non-thermal plasma sources analyzed must be added.

Author Response

  1. The authors present a review on the recent progress of atmospheric cold plasma in food processing within the past decade. Current challenges as well as its future outlook are also presented. However, the description of the non-equilibrium plasmas as well as the plasma-based devices contain several errors and should be considerably improved before suggesting the manuscript for publication. Please note that a non-equilibrium or cold plasma is not a subcategory of non-thermal plasma. The terms non-thermal plasma, cold plasma and non-equilibrium plasma refer all to a state of plasma which is far from the local thermodynamic equilibrium. Particularly the non-thermal plasmas with gas temperatures below 60 °C are suitable for plasma medicine and biological applications (as food processing). In addition, the sentence ‘In order to generate cold plasma, a low-gas pressure system is required to minimize molecular collisions’ should be explained or removed.

Response: Sorry for the confusion and thank you for your kind reminder. This sentence has been modified, please refer to lines 48-51.

  1. Also, the sentence ‘DBD offer certain advantages over other plasma devices: Principally, high energy species can be generated from DBD with as minimal energy as 1 eV’ is wrong, and should be rewritten or removed.

Response: Sorry for the confusion and thank you for your kind reminder. This sentence has been modified, please refer to lines 99-101.

  1. The Section 2 Mechanism of ACP needs to be rewritten. Typical plasma values expected for the non-thermal plasma sources analyzed must be added.

Response:The mechanism of ACP has been rewritten to provide more clarity, and typical plasma values for the non-thermal plasma sources analyzed will be added. Section 2 has been modified and we apologize for any confusion caused. Please see the Section 2.

Reviewer 2 Report

The article is interesting, well organized and clear in its writing.

As an observation the authors could improve the part where they describe the different technologies for the generation of cold plasma, they only develop the corona discharge and the plasma jet without describing the plasma generation by RF, spark and glow discharge, gliding arc, and SBD.

Author Response

The article is interesting, well organized and clear in its writing. As an observation the authors could improve the part where they describe the different technologies for the generation of cold plasma, they only develop the corona discharge and the plasma jet without describing the plasma generation by RF, spark and glow discharge, gliding arc, and SBD.

Response: Thank for your kind comments. The mechanism of ACP has been rewritten to provide more clarity, and typical plasma values for the non-thermal plasma sources analyzed will be added. Section 2 has been modified and we apologize for any confusion caused. Please see the Section 2.

Round 2

Reviewer 1 Report

The authors have reasonably responded to the points raised.